# Comparison of Wuqinxi Qigong with Stretching on Single- and Dual-Task Gait, Motor Symptoms and Quality of Life in Parkinson’s Disease: A Preliminary Randomized Control Study

**DOI:** 10.3390/ijerph19138042

**Published:** 2022-06-30

**Authors:** Zhenlan Li, Tian Wang, Mengyue Shen, Tao Song, Jie He, Wei Guo, Zhen Wang, Jie Zhuang

**Affiliations:** 1School of Sport Science, Shanghai University of Sport, Shanghai 200438, China; cony2649@sina.com (Z.L.); wangtiann@126.com (T.W.); 609st@163.com (T.S.); hejie970529@163.com (J.H.); 2Department of Rehabilitation Sciences, Ningbo College of Health Sciences, Ningbo 315100, China; 3School of Martial Arts, Shanghai University of Sport, Shanghai 200438, China; s15978779392@126.com (M.S.); 15670586597@163.com (W.G.); wangzhen@sus.edu.cn (Z.W.); 4School of Physical Education, Jianghan University, Wuhan 430056, China

**Keywords:** Wuqinxi Qigong, stretching, dual-task gait, motor symptoms, Parkinson’s disease

## Abstract

The objective of this study was to investigate the therapeutic effect of Wuqinxi Qigong vs. stretching on single- and dual-task gait, motor symptoms, and quality of life in people with mild and moderate Parkinson’s disease (PD). This single-blind, randomized control trial included 40 participants with idiopathic PD who were randomized into the Wuqinxi Qigong (WQ) group or stretching group. Participants completed 12 weeks (two sessions/week) of intervention. The primary outcomes were gait parameters when performing single-task (comfortable pace) and dual-task (obstacle crossing, serial-3 subtraction and backward digit span) walking, including gait speed, stride length, and double support percentage. The secondary outcomes were ratings from the Movement Disorder Society Unified Parkinson’s Disease Rating Scale (MDS-UPDRS), results of the timed-up-and-go test (TUGT), results of the Mini-Balance Evaluation Systems Test (MiniBESTest), and responses from the 39-item Parkinson’s Disease Questionnaire (PDQ-39). All measures were assessed pre- and post-intervention. The WQ group demonstrated increased gait speed (*p* = 0.000) during the single task, and increased stride length (*p* = 0.001, *p* = 0.021) during the single-task and serial-3 subtraction task. Double support percentage significantly decreased (*p* = 0.004) in the WQ group during the obstacle crossing task, and also decreased (*p* = 0.045) in the stretching group during the single-task. TUGT (*p* = 0.005), MiniBESTest (*p* = 0.023) and PDQ-39 (*p* = 0.043) in the WQ group significantly improved, and both groups showed significant improvement in MDS-UPDRS after intervention. Wuqinxi Qigong is an effective method to improve single- and dual-task gait. While both exercises improve motor symptoms, Wuqinxi Qigong results in better mobility, balance and quality of life compared to stretching alone.

## 1. Introduction

Parkinson’s disease (PD) is a chronic neurodegenerative disorder that afflicts many patients globally. Gait impairment, characterized by short, asymmetric steps and slow speed, is a main motor symptom in patients with PD. As the disease progresses, gait function decrements increase the risk of falls [1]. Locomotor tasks often require acute attention and processing speed [2], and most daily activities depend on the simultaneous functioning of motor and cognitive tasks. Indeed, “normal” behaviors demand continuous integration of neuronal processing and the “dual-task” (DT) of cognitive and motor functioning [3]. However, gait deficits in patients with PD worsened under dual-task conditions and seriously affect walking ability [4,5]. When performing dual-tasks, people with Parkinson’s disease may concentrate on a secondary task (e.g., walking while speaking) and pay less attention to walking [6,7], resulting in reduced walking speed and stride length, increased gait variability, and subsequent increased risk of falling [8].

Although dual-task gait deficits cannot be treated with dopamine drugs [9], targeted motor or cognitive dual-task exercise can improve dual-task gait and balance [10]. Accumulating evidence suggests that the pattern of dual-task training improves gait and postural instability in people with Parkinson’s disease. The current research found that six consecutive weeks of integrated cognitive and gait task training in individuals with PD remarkably increased stride length and cadence [11]. Furthermore, adding cognitive training to a treadmill program alters the effects of training on the magnitude and lateralization of prefrontal lobe activation to reduce fall risk [12]. Preliminary data suggest 12 sessions of cognitive dual-task training (CDTT) reduces time of double support during cognitive dual-tasks; motor dual-task training (MDTT) further decreases gait variability during motor dual-tasks [11]. Notably, these studies combined two different tasks to improve the gait function; however, whether one type of exercise therapy is sufficient to obtain similar results remains unclear.

Traditional Chinese medicinal exercise is a method that integrates body movement with mental focus. Among the traditional Chinese medicinal exercise, Tai Chi and Qigong are the most practiced, and both aid in health and eliminate disease by incorporating body movement, mindful breathing, and cognitive training [13]. In contrast to Tai Chi, Qigong emphasizes standing posture and dynamic movement. Qigong combines position adjustment, breathing exercises and mindfulness practice to activate muscles and tendons [14], which promotes lower limb function and improves dual-task gait associated with cognition [15,16].

Huato, a physician in the Donghan Dynasty of ancient China, first developed the Wuqingxi method of Qigong exercise. It has since been edited by the Chinese Health Qigong Association and adopts the movement of five animals, including tiger, deer, bear, ape and bird [17]. Importantly, Wuqinxi improves health-related parameters in both healthy individuals and patients with chronic disease (i.e., chronic obstructive pulmonary disease and emphysema) [17,18]. Previous studies suggest that Wuqinxi is an effective at-home exercise that can alleviate chronic low back pain by improving the function of lumbosacral multifidus and reduce knee pain by enhancing knee muscle strength [19]; this has been particularly beneficial for elderly patients with knee osteoarthritis [20,21]. These studies demonstrate the effectiveness of Wuqinxi Qigong in helping respiratory, musculoskeletal and cognitive functions. However, the benefits of Wuqinxi Qigong on dual-task gait, balance and other functions remains unknown.

Therefore, we adopted a randomized control trial to compare the influence of Wuqinxi Qigong and stretching on gait, motor symptoms and quality of life in PD. The primary goal was to explore the differential effects of Wuqinxi Qigong versus stretching alone on single- and dual-task gait. The secondary objective was to examine their impact on balance function, mobility, and quality of life for patients with PD.

## 2. Materials and Methods

### 2.1. Study Design and Participants

This prospective, single-center, single-blind randomized controlled study compared the effects of Wuqinxi Qigong (WQ) versus stretching for individuals with PD. A total sample of 100 participants was approved by the ethics committee of Shanghai University of Sport and was registered online at the Chinese Clinical Trial Registry (ChiCTR1800016570). Participants were recruited from referrals by the neurology department of University School of Medicine and from the local community. A research assistant contacted participants interested in the study to lower potential expectation bias and confirm eligibility. Inclusion criteria were: (1) aged 60–80 years; (2) Idiopathic Parkinson’s disease according to British brain bank criteria [22]; (3) Hoehn and Yahr (H&Y) stage of 1–3; (4) Mini-Mental State Examination score ≥ 24; (5) ability to walk independently for at least 5 min (walking AIDS were allowed). Exclusion criteria: (1) experience practicing Wuqinxi Qigong; (2) severe cognitive, visual or auditory impairment, or unable to understand instructions and express demands; (3) other diseases such as cardiopulmonary diseases, liver and kidney diseases, tumors, musculoskeletal dysfunction, and any disease that may limit exercise; (4) irregular use of Parkinson’s disease medication. All participants were required to provide signed informed consent prior to baseline assessment. This study complies with the CONSORT guidelines (see Appendix A).

### 2.2. Sample Size Calculation and Randomization

This RCT study aimed to determine whether Wuqinxi Qigong is superior to stretching for gait performance. In the absence of preliminary data on the effects of Wuqinxi Qigong and stretching on dual-task walking, we estimated the effect size in power calculations from previous studies that compared different patterns of dual-task gait training [23]. A sample size of 40 was determined after a power calculation utilizing G*Power software (version 3.1.9.3, Düsseldorf, Germany) was performed. This calculation was based on between group differences in primary outcomes (dual-tasking gait performance) between the Wuqinxi Qigong group and stretching group. Applying an effect size of 0.8, a type I error of 0.05, and 70% power, and accounting for a 15% attrition rate, at least 20 participants in one group needed to reach a statistically significant difference in the dual-task gait performance. Eligible participants were randomized to one of the interventions with an allocation of 1:1 through a permuted bloc randomization (STATA V.12.0, Stata Corp, Texas, USA). The random number table was generated by the program data analyst, and handled exclusively by a research staff member who gave it to an assistant in a sealed envelope. The assistant then notified participants of their intervention group. Assessors and investigators were blind to the expected study outcome.

### 2.3. Interventions

Both the WQ group and stretching group received group-based exercise at the University’s laboratory of sports science, and were asked to abstain from any extra at-home exercises throughout the twelve-week training period. Groups contained eight to ten participants in an effort to provide adequate instruction and attention to each. Two weekly 90-min sessions were conducted for 12 consecutive weeks. All exercise sessions were supervised at the study site. Each session included a 10-min warm-up on breathing and joint exercise, 60 min with Wuqinxi or stretching, 10 min of break intervals and a 10-min cool-down to relax muscle groups. Intensity was calculated by measuring heart rate (polar-team2; Polar Electro Finland) and having participants rate their perceived exertion (RPE). The heart rate at the start and end of the training session was set at 30–39% of the maximum heart rate (HR_max_) for age, or at RPE of 6 on Borg scale. The training intensity was equivalent to 60~70% of HR_max_, determined by the formula HR_max_ = 208 − (0.7 × age), or determined by whether participants reaches their limit of fatigue at an RPE of 12. During the first 6 weeks, participants were instructed to perform at a level of exertion that they felt “Light to Moderate” (equivalent to 6–9 on Brog scale) and developed to “Moderate to Strong” and “Strong” (equivalent to 10–12 on Brog scale) after 6 weeks of intervention [24]. All participants wrote their intervention-based activity in an exercise log throughout the trial.

#### 2.3.1. Wuqinxi Qigong Exercise

Wuqinxi Qigong consists of 10 forms: (1) Raising the tiger’s paws, (2) Seizing the prey, (3) Colliding with the antlers, (4) Running like a deer, (5) Rotating the waist like a bear, (6) Swaying like a bear, (7) Lifting the monkey’s paws, (8) Picking fruit, (9) Stretching upward and (10) Flying like a bird (see Figure 1 and Appendix A) [25]. During the first 2–3 weeks, exercise primarily emphasized learning and practicing two or three of these movements. Participants were instructed to repeat each movement six times and to incorporate natural breathing into the movement routine. Although exercise was progressive, movement pace, pattern and coordination, and joint range of motion were adjusted depending on whether each participant could keep up with the exercise. The instructor gave instructions and demonstrations, requiring participants to count or recite each form of Wuqinxi while practicing, as well as focus on maintaining balance, posture and action consistency.

#### 2.3.2. Stretching Exercise

Participants in the stretching group were instructed to practice stretching in both sitting and standing positions. Stretching is beneficial for relaxing stiff muscles and increasing joint flexibility. The stretching methods utilized by this group included self-stretching targeted toward upper limb muscles (deltoid, biceps, triceps), low limb muscles (iliopsoas, quadriceps, hamstring, tibialis anterior, gastrocnemius), and trunk muscles (pectoralis major, erector spinae) (see Appendix A) [26]. Each muscle group was activated three to four times for 30 s each, with a 10 s resting period between each stretch. Participants were guided to breathe naturally and intensify the stretch to mild discomfort.

### 2.4. Outcome Assessments

All participants received assessments before and after the 12-week intervention. Baseline measurements included age, gender, cognition, body mass index (BMI), duration of disease, Hoehn and Yahr phase and freezing of gait. Participants performed each assessment at the “on” medication phase before and after the intervention. All experimental data were collected by the same trained assessor to ensure reproducibility.

The primary outcomes were gait parameters, measured using a 4.6 m long electronic walkway carpet with embedded pressure sensors (PKMAS walkway, ProtoKinetics, Havertown, PA, USA). Participants completed four types of tasks while walking on walkway: (1) single-task walking (comfortable pace); (2) obstacle crossing (15 cm height); (3) backward digital span; and (4) serial-3 subtraction. Gait parameters included gait speed, stride length and double support percentage during single and dual tasks. To avoid a learning effect, the order of each task changed randomly. Participants walked three consecutive loops on the walkway to minimize within-individual variability [27].

The secondary outcomes were motor symptoms, balance, mobility, and quality of life. Motor symptoms were rated by the Movement Disorder Society Unified Parkinson’s Rating Scale (MDS-UPDRS) [28]. Balance and mobility performance were assessed by Mini-Balance Evaluation Systems Test (MiniBESTest) [29] and Time-Up and Go Test (TUGT) [30]. Mini-Mental State Examination measured cognition [31]. Finally, quality of life was rated using the 39-item Parkinson’s Disease Questionnaire (PDQ-39) [32].

### 2.5. Statistical Analysis

Statistical analyses were conducted using SPSS V.22.0 software (IBM Corp, New York, NY, USA). The Shapiro-Wilk test determined data normality. Baseline demographics were examined using chi-square tests. Independent *t*-tests were utilized to analyze group differences. Significant between-group differences were determined using analysis of covariance. Variance homogeneity of data was assessed by Levene’s test. The outcome of the normally distributed data before (Pre) and after 12-week intervention (Post_12wk_) was analyzed using mixed-design repeated measures ANOVA for between/within-group differences. When an interaction effect was determined, one-way repeated-measures ANOVA was utilized to compare the within-group differences. Bonferroni’s post hoc test was then applied to compare results when main effects were significant. All descriptive statistics are presented as mean and standard deviation; significance level was set at *p* < 0.05. Intention-to-treat principles were used to address missing data, that is, all individuals initially enrolled in the study. Effect size was calculated using partial eta-squared (η^2^). The effect size thresholds were small = 0.01, medium = 0.06, large = 0.14 [33].

## 3. Results

### 3.1. Participants and Baseline Characteristics

Forty participants were enrolled and randomly divided into the WQ group (*n* = 20) or stretching group (*n* = 20). One participant in each group withdrew due to medical issues unrelated to PD and did not complete the 12-week intervention, for an attrition rate of 5%. All the participants were analyzed at the end of the study (see Figure 2). No adverse events (such as fall, syncope or sports injury) occurred during intervention. Baseline characteristics showed no statistical differences between the two groups in age, gender, BMI, duration of disease, education, Hoehn and Yahr stage, motor function, or cognitive function (see Table 1).

### 3.2. Intervention Effects in Primary and Secondary Outcomes

Table 2 and Table 3 demonstrate the training effects on primary and secondary outcomes following the 12-week intervention. The WQ group demonstrated significantly increased performance from baseline in gait speed (*p* = 0.045) and stride length (*p* = 0.014) compared to the stretching group at post_12wk_. Between-group analysis demonstrated that the changes of gait speed in the stretching group from baseline is higher than the WQ group among the obstacle crossing task (*p* = 0.041). The changes of double support percentage in both groups during all dual-task measures showed a significant difference (*p* < 0.05), and the decrease of double support percentage in the WQ group was greater than that in stretching group. Furthermore, participants in the WQ group showed significantly increased gait speed (increase of 6.39 ± 5.98 cm/s; *p* = 0.000) and stride length (increase of 5.85 ± 6.45 cm; *p* = 0.001) following 12 weeks of intervention. Stride length also increased by 5.04 ± 8.98 cm (*p* = 0.021) in the serial-3 subtraction task. Double support percentage in the stretching group (from 33.27 ± 7.79 to 31.36 ± 8.19; *p* = 0.045) and WQ group (from 26.19 ± 4.59 to 24.15 ± 3.60; *p* = 0.04) was remarkably reduced as a single-task and obstacle-crossing task.

The MDS-UPDRS Part III and total MDS-UPDRS scores in the WQ group significantly improved after 12 weeks of training (*p* = 0.021, *p* = 0.004), and decreased by 8.68 ± 14.96 points and 18.44 ± 23.34 points, respectively. The stretching group decreased on the MDS-UPDRS scores by 7.60 ± 13.34 points (*p* = 0.020). In the WQ group, TUGT time increased from 11.43 ± 3.48 to 10.05 ± 2.01 (*p* = 0.005), and MiniBESTest scores increased from 20.05 ± 7.18 points to 24.52 ± 4.20 points (*p* = 0.023). Finally, the PDQ-39 score significantly decreased from 30.00 ± 17.24 points to 22.37 ± 15.88 points (*p* = 0.043). These variables had no significant changes in the stretching group.

## 4. Discussion

We report that WQ improves walking speed and stride length of PD patients in the single and serial-3 subtraction tasks. Wuqinxi Qigong also reduces the double support percentage when participants cross obstacles, while stretching exercise reduces the double support percentage during single-task walking. Both exercise paradigms significantly alleviate motor symptoms, although WQ improves balance, mobility and quality of life better than stretching alone.

The gait characteristic of Parkinson’s disease is a bent posture; a decline in arm swing, gait speed, and stride length; and a prolonged double support phase which tends to cause falls [9,34]. The gait speed of participants in the WQ group increased by 0.0639 m/s, a value higher than the smallest meaningful change estimates (ranging from 0.04 to 0.06 m/s for gait speed) in older adults [35]. The 5.85 cm increase in stride length following 12-weeks of intervention demonstrates that WQ improves gait speed and stride length in single-task walking. Notably, several forms of Wuqinxi, such as swaying like a bear and picking fruit, require participants to move and change the center of gravity. This, accompanied with the movement of the upper limbs, slow movement, trunk control in space and shifting center of gravity, may account for the increase in gait stride. The movement patterns in WQ may benefit one’s ability to concentrate on supporting surface and postural stability, as well as ameliorate stride length and decline walking impairment [36,37].

This study showed that the double support percentage reduced when participants in the WQ group performed the obstacle-crossing task. Dual-task gait is related to motor and cognitive function, and the obstacle-crossing task is also related to muscle strength, executive function, attention and visual special abilities. Patients with Parkinson’s disease tend to increase their support base, reduce gait speed, and prolong double support time to prevent the risk of trips and falls when approaching obstacles [38,39]. Here, double support percentage is expressed as the percentage of total double support time to gait cycle time. The double support time is an indicator related to dynamic postural stability, which is associated with changes in stride length and stride time [40,41]. The current study shows that Wuqinxi Qigong intervention significantly reduced stride time variability (by 36.2%) and stride length variability (by 17.5%) for PD patients [34]. Importantly, Wuqinxi Qigong exercise requires participants to move and change the center of gravity following movements of the upper limbs. Slow movements, trunk control in space, and the transfer of the center of gravity in different directions was also emphasized. Moreover, most Qigong movements are closed-chain exercises of the lower limbs, and thus conducive to correction of inadequate heel stride and knee extension on gait cycle [42,43].

Previous studies have shown that increased attention demands, problem solving, recalling a memory, or engaging in visuospatial processing while walking in patients with Parkinson’s disease can lead to gait deterioration, cognitive decline, or both [44,45]. Dual-task training decreases time of double support during cognitive dual-task walking and reduces variability of gait during motor dual-task walking; it also improves the gait speed and stride length and reduces double support time in both motor dual-task and single-task walking [46]. Our study is consistent with these results and demonstrates that a 12-week WQ exercise regimen increases stride length in the serial-3 subtraction task, reduces double support percentage in the obstacle-crossing task, and improves gait speed and stride length in single-task walking. Wuqinxi Qigong includes cognitive components, and participants memorized the name of each form when performing the exercise. Each participant imitated and learned every movement and was able to practice all movements independently. This might increase cognitive functions such as memory, visuospatial, executive, and retelling. Moreover, we found that stretching exercise reduces double support percentage in a single-task. Several studies have demonstrated that static stretching combined with lower muscle strength improves balance and prevents falls in the elderly. It also improves motor symptoms and mobility and increases the speed of walking backwards for patients with PD [47,48].

Our study showed that TUGT increased from 11.43 ± 3.48 s to 10.05 ± 2.01 s, indicating an improvement in mobility after practicing WQ. Furthermore, the MiniBESTest score was 4.48 points higher than pre-intervention levels in the WQ group, which demonstrates a significant improvement in balance function. Notably, these levels reached the minimal clinically important difference of the MiniBESTest of 4 points [49]. The total score of the MDS-UPDRS test in the WQ and stretching exercise groups decreased by 18.44 ± 23.34 and 7.60 ± 13.43 points, respectively, which exceeded the threshold of 3.5 points for clinically important MDS-UPDRS score differences [50] These results suggest that both WQ and stretching improve motor symptoms of Parkinson’s patients and delay the progression of disease. Finally, the PDQ-39 scores in the WQ group decreased significantly, indicating that 12 weeks of WQ exercise effectively improves quality of life.

There are several limitations to this study. First, it was not feasible for participants and investigators to achieve a doubled-blind status for this study because intervention exercises were extensively open to the participants. Second, the results are only generalizable to patients with mild to moderate PD; whether these methods work similarly for advanced PD remains unknown. Third, intervention duration was relatively short and follow-up after intervention was not performed to evaluate long-term effects. Fourth, the study had an insufficient sample size; the addition of other centers to this trial as well as participants from other geographic areas would further strengthen these results. Future investigations should aim to increase the sample size and investigate the long-term therapeutic effects of WQ on patients with PD.

## 5. Conclusions

This study demonstrates that Wuqinxi Qigong improves single- and dual-task gait. While both Wuqinxi Qigong and stretching improve motor symptoms, Wuqinxi Qigong also improves mobility, balance, and quality of life to a greater extent than stretching alone. Additional investigation is required to explore the long-term sustained effects of Wuqinxi Qigong intervention in individuals with PD.

## Figures and Tables

**Figure 1 ijerph-19-08042-f001:**
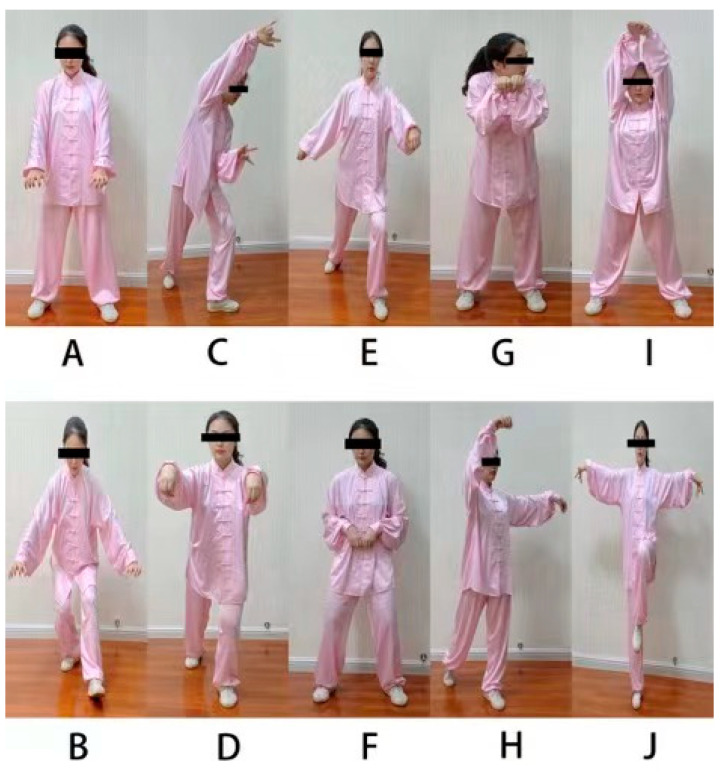
Ten forms of Wuqinxi Qigong. (**A**). Form 1: Raising the tiger’s paws; (**B**). Form 2: Seizing the prey; (**C**). Form 3: Colliding with the antlers; (**D**). Form 4: Running like a deer; (**E**). Form 5: Rotating the waist like a bear; (**F**). Form 6: Swaying like a bear; (**G**). Form 7: Lifting the monkey’s paws; (**H**). Form 8: Picking fruit; (**I**). Form 9: Stretching upward; (**J**). Form 10: Flying like a bird.

**Figure 2 ijerph-19-08042-f002:**
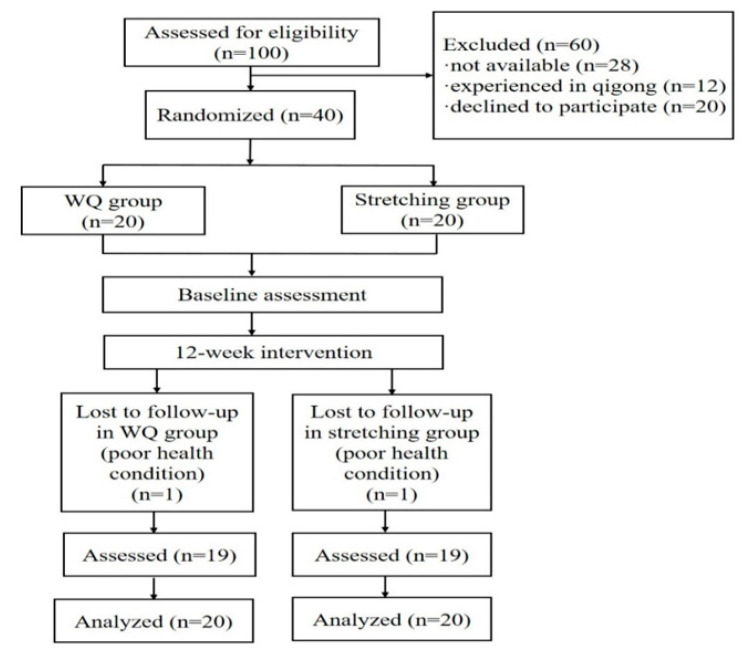
Flow diagram of the study.

**Table 1 ijerph-19-08042-t001:** Demographic characteristics and clinical outcomes.

Demographics and Outcomes	WQ Group (*n* = 20)	Stretching Group (*n* = 20)	*p*
Age (year)	67.57 ± 3.95	70 ± 5.59	0.126
Gender (M/F)	13/7	16/4	0.288
Education condition (years)	14.40 ± 1.47	14.55 ± 1.36	0.739
Disease duration (years)	6.83 ± 4.09	7.76 ± 4.55	0.528
H&Y stage, *n* (%)			0.631
1–1.5	10 (50%)	7 (35%)	
2–2.5	7 (35%)	9 (45%)	
3	3 (15%)	4 (20%)	
Freezing of gait, *n* (%)	9 (45%)	9 (45%)	1.000
MDS-UPDRS	25.05 ± 17.45	26.79 ± 20.79	0.598
MDS-UPDRSIII	52.19 ± 26.87	47.47 ± 29.25	0.646
MMSE	26.79 ± 2.58	28.43 ± 2.58	0.157
Falls in the 6 months prior to study, *n* (%)	3 (15%)	4 (20%)	1.000
LEDD (mg/d)	450 ± 252.98	450 ± 164.99	1.000

Note. MDS-UPDRS: Movement Disorder Society-the Unified Parkinson’s Disease Rating Scale; H&Y: Hoehn and Yahr stage; LEDD: levodopa equivalent daily dose; MMSE: Mini-Mental State Examination; Education condition: primary school-six years, junior high school-three years, senior high school-three years, college-four or three years; WQ: Wuqinxi Qigong.

**Table 2 ijerph-19-08042-t002:** Within-group comparisons of clinical outcomes.

Outcomes	Variables	WQ Group			Stretching Group			Time * Group
		Pre	Post_12wk_	Pre vs. Post_12wk_ *p*	Pre	Post_12wk_	Pre vs. Post_12wk_ *p*	*p*
*Primary* *outcomes*								
Single-task	Gait speed (cm/s)	91.03 ± 9.34	97.42 ± 10.85	0.000 *	79.85 ± 10.49	80.63 ± 13.12	0.650	0.964
	Stride length (cm)	101.92 ± 11.57	107.78 ± 10.57	0.001 *	95.58 ± 11.45	95.92 ± 11.59	0.118	0.001 *
	Double support%	28.83 ± 5.66	28.63 ± 5.87	0.548	33.27 ± 7.79	31.36 ± 8.19	0.045 *	0.031 *
Obstacle crossing	Gait speed (cm/s)	89.29 ± 13.78	89.37 ± 17.67	0.978	74.14 ± 24.52	78.53 ± 23.96	0.212	0.330
	Stride length (cm)	98.21 ± 13.26	102.32 ± 12.29	0.145	92.83 ± 28.56	90.05 ± 27.79	0.317	0.080
	Double support%	26.19 ± 4.59	24.15 ± 3.60	0.004 *	29.45 ± 9.79	30.79 ± 10.19	0.259	0.012 *
Backward digit span	Gait speed (cm/s)	77.35 ± 20.95	75.23 ± 15.41	0.524	59.86 ± 23.95	60.55 ± 26.59	0.783	0.500
	Stride length (cm)	85.73 ± 15.36	85.55 ± 12.17	0.951	74.97 ± 26.99	73.61 ± 27.47	0.608	0.754
	Double support%	32.11 ± 5.60	31.49 ± 5.22	0.596	37.49 ± 11.45	38.64 ± 10.93	0.251	0.250
Serial-3 subtraction	Gait speed (cm/s)	64.99 ± 18.12	66.52 ± 13.84	0.544	56.19 ± 21.61	59.64 ± 24.29	0.291	0.633
	Stride length (cm)	77.69 ± 13.03	82.73 ± 10.39	0.021 *	71.47 ± 25.30	73.09 ± 26.11	0.587	0.336
	Double support%	34.27 ± 7.33	32.13 ± 5.71	0.171	39.23 ± 11.78	38.49 ± 10.75	0.593	0.494
*Secondary outcomes*								
	MDS-UPDRSIII	21.63 ± 18.36	12.95 ± 9.00	0.021 *	24.15 ± 15.89	21.55 ± 16.50	0.051	0.098
	MDS-UPDRS	43.78 ± 29.95	25.33 ± 16.38	0.004 *	46.55 ± 28.77	38.95 ± 32.17	0.020 *	0.084
	TUGT (s)	11.43 ± 3.48	10.05 ± 2.01	0.005 *	11.24 ± 3.90	13.19 ± 4.48	0.029	0.001 *
	MiniBESTest	20.05 ± 7.18	24.52 ± 4.20	0.023 *	22.15 ± 5.88	22.35 ± 5.61	0.779	0.038 *
	PDQ-39	30.00 ± 17.24	22.37 ± 15.88	0.043 *	26.79 ± 20.29	22.42 ± 18.87	0.166	0.486

Note. MDS-UPDRS, Movement Disorder Society-the Unified Parkinson’s Disease Rating Scale; MiniBESTest, Mini-Balance Evaluation Systems Test; PDQ-39, 39-item Parkinson’s Disease Questionnaire; TUG, timed-up-and-go test; WQ, Wuqinxi Qigong; * *p* < 0.05.

**Table 3 ijerph-19-08042-t003:** Between-group comparisons of changes from baseline for clinical outcomes.

Outcomes	Variables	Pre vs. Post_12wk_			
		WQ Group	Stretching Group	*p*	Effect Size
*Primary outcomes*					
Single-task	Gait speed (cm/s)	−6.39 ± 5.98	−0.79 ± 7.43	0.045 *	0.105
	Stride length (cm)	−5.85 ± 6.49	−0.34 ± 0.90	0.014 *	0.152
	Double support%	0.20 ± 1.49	1.53 ± 3.12	0.135	0.059
Obstacle crossing	Gait speed (cm/s)	−0.08 ± 12.42	−4.39 ± 14.76	0.041 *	0.108
	Stride length (cm)	−4.11 ± 12.09	2.78 ± 11.78	0.195	0.045
	Double support%	2.05 ± 2.78	−1.34 ± 5.02	0.041 *	0.108
Backward digit span	Gait speed (cm/s)	2.12 ± 14.62	−0.70 ± 10.89	0.862	0.001
	Stride length (cm)	0.17 ± 12.17	1.36 ± 11.37	0.094	0.074
	Double support%	0.62 ± 5.15	−1.15 ± 4.22	0.025 *	0.128
Serial-3 subtraction	Gait speed (cm/s)	−1.52 ± 11.01	−3.44 ± 13.81	0.200	0.044
	Stride length (cm)	−5.04 ± 8.98	−1.62 ± 12.76	0.201	0.044
	Double support%	2.14 ± 6.74	0.74 ± 5.91	0.047 *	0.103
*Secondary outcomes*					
	MDS-UPDRSIII	8.68 ± 14.96	2.60 ± 5.58	0.165	0.053
	MDS-UPDRS	18.44 ± 23.34	7.60 ± 13.43	0.339	0.025
	TUGT (s)	1.38 ± 1.98	−1.95 ± 3.71	0.155	0.051
	MiniBESTest	−4.48 ± 8.33	−0.20 ± 3.14	0.981	0.000
	PDQ-39	7.63 ± 15.30	4.37 ± 13.16	0.774	0.002

Note. MDS-UPDRS, Movement Disorder Society-the Unified Parkinson’s Disease Rating Scale; MiniBESTest, Mini-Balance Evaluation Systems Test; PDQ-39, 39-item Parkinson’s Disease Questionnaire; TUG, timed-up-and-go test; WQ, Wuqinxi Qigong; * *p* < 0.05.

## Data Availability

The data presented in this study are available on request from the corresponding author.

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
