# Peer review of "Comparison of Wuqinxi Qigong with Stretching on Single- and Dual-Task Gait, Motor Symptoms and Quality of Life in Parkinson’s Disease: A Preliminary Randomized Control Study"

_ijerph, 2022, doi:10.3390/ijerph19138042_

Round 1

Reviewer 1 Report

This is a very well written and methodologically correct article; I will made just some small suggestion on the use of the repeated measures mixed ANOVA: what type of effect size has been used? the most correct is partial square omega because it eliminates the bias in this type of design.  Why use the Bonferroni correction? (I suggest reporting Levene's test). It would also be good to report the result of Shapiro-Wilk test of normality (it only takes one line)

Author Response

Response 1:

We revised the description of Statistical methods. (see Line 223-229)

1.We added the type of effect size according to your advice, using partial eta-squared (η2).

2.The variance homogeneity of data was assessd by Levene's test, which which is a precondition for parametric tests such as the t-test and ANOVA. Bonferroni’s correction is imperative to avoid a type I error, especially the number of tests increases. It has become a popular method and is widely used in various experimental contexts including: (1) comparing different groups at baseline, (2) studying the relationship between variables, and (3) examining more than one endpoint in clinical trials. (Armstrong RA. When to use the Bonferroni correction. Ophthalmic Physiol Opt 2014; 34: 502–508.)

Reviewer 2 Report

Thank you for letting me review this interesting manuscript about Comparison of Wuqinxi Qigong with Stretching in patients with Parkinson. 

I have serious concerns about the methods of the manuscript that must be solve to go further: 

Line 104-108. Please describe the number of the Ethical Committe approval. This information must appear at the begining of the material and methods section. No below the elegibility criteria. 

The figure 1 is not the CONSORT statement. Is the flowchart diagram. So, it shoyud be included in the result paragraph.

CONSORT guidelines should be reported as Supplementary data. 

Line 124-125. The sample size calculation is not correct. The authors shoudl explain if the RCT was a superiority or non-inferiority study. Otherwise, when a comparison between two techniques is carried out, the infromation needed is between-groups instead of within groups

Line 124. Is the previous study avaliable for reading? 

Line 127-130. Who carried out the randomization? 

Line 138-139. How was the initial intensity calculated and how was the progression made? 
The same intensity was applied to all the participants? What type of exercises did you include? Further details of this paragraph are needed in order to replicate it. 

Line 148-153. How was adjusted? Which was the initial level of intensity and how was calculated?
More information is needed. 

Line 150-151. How did the interventor do that? 

Line 155-162. References?

abbreviations should be checked throughout the manuscript

Author Response

Response 1: We added the number of the Ethical Committe approval at the beginning of the material and methods section (see Line 99-101).

Response 2: We put our flow diagram in the result paragraph according to your advice,

and added the checklist of CONSORT (see Supplementary File).

Response 3: We added the CONSORT guideline as Supplementary data.

Response 4: We revised sample size calculation according to your advice. We searched the reference on dual-task training to add some information about sample size calculation. (see Line 122-132)

Response 5: Xiao, C,M.; Zhuang, Y,C. Effect of health Baduanjin Qigong for mild to moderate Parkinson's disease. Geriatr Gerontol Int. 2015, 16, 911-919.

This study reported the significant difference in gait performance based on within-group. We have replaced this reference.

Response 6: Eligible participants were randomly to one of the interventions with an allocation of 1:1 through a permuted block randomization. The randomization digit table was generated by the program data analyst (see Line 135-139).

Response 7: Initial intensity was calculated using by heart rate, and participants used heart rate monitor (polar-team2; Polar Electro, Finland) and rating of perceived exertion to record intensity of all exercise sessions. Exercise frequency and intensity were adjusted based on individual heart rate data, until reached the target level. Each session included a 10-minute warm-up on breathing and joints exercise, 60 minutes with Wuqinxi or stretching exercise, 10 minutes of break intervals and a 10-minute cool-down on relaxation of muscle groups (see Line145-151).

Response 8: Initial intensity was calculated using by heart rate monitor. We added the information that how to adjust the intensity and exercise. Movement pace, pattern and coordination, and joint range of motion were are adjusted depending on whether each participants can follow the exercise.(see Line 164-169)

Response 9: The instructor gave instructions and demonstrations, requiring participants to count or recite each form of Wuqinxi while practicing (see Line 176-19).

Response10: We added related reference. Bob Anderson. Stretching: 40th Anniversary Edition. Shelter Publication Inc., U.S, 2020.

Response11: We corrected the abbreviations throughout the manuscript.

Reviewer 3 Report

Dear authors:

The paper "Comparison of Wuqinxi Qigong with Stretching on Single- and 2 Dual-task Gait, Motor symptoms and Quality of Life in Parkin-3 son’s Disease: A Preliminary Randomized Control study"

Thank you very much for your research.

I consider it is a good paper, well written and very clear and it can be understood by the reader, is the most important point.

But I want to ask you the figure inclusion about the uqinxi Qigong exercise. This lets the readers understand both comparations of different types of exercises, 

Thank you very much

Author Response

Response: We have made a detailed description of the movements of Wuqinxi and stretching, please see the Supplement File.

Round 2

Reviewer 2 Report

The authors have answered all my questions. However, one question is not completed. 

THe authors replied: "Initial intensity was calculated using by heart rate, and participants used heart rate monitor (polar-team2; Polar Electro, Finland) and rating of perceived exertion to record intensity of all exercise sessions. Exercise frequency and intensity were adjusted based on individual heart rate data, until reached the target level.". 

Which was the initial heart rate and RPE? which was the target level at the begining and at the end? 

Author Response

Dear Reviewer,

We would like to thank you to give us comments for our manuscript (ID ijerph-1671744) again. Your comments are all valuable and helpful for revising and perfecting our paper. We have carefully revised the manuscript according to your questions.

In the revised version, changes to our manuscript were all highlighted within the document by using red and blue colored text. Point-by-point responses to your comments are listed below this letter.

Response: The heart rate at the start and end of the session was set at 30%–39% of the maximum heart rate (HRmax) for age, or at RPE of 6 on the Borg scale. The target level reached 60%~70% of HRmax, determined by the formula HRmax = 208-(0.7 × age), or determined by whether participants reaches their limit of fatigue at an RPE of 12. (see revised version Line 155-159)

Thank you for your help.

Best wishes,

Zhenlan Li
